# Damage Resistance of Kevlar^®^ Fabric, UHMWPE, PVB Multilayers Subjected to Concentrated Drop-Weight Impact

**DOI:** 10.3390/polym16121693

**Published:** 2024-06-14

**Authors:** Manal A. Nael, Dmitriy A. Dikin, Natnael Admassu, Omar Bahgat Elfishi, Simona Percec

**Affiliations:** 1Chemistry Department, College of Science and Technology, Temple University, 1925 N 12th Street, Philadelphia, PA 19122, USA; manal.nael@temple.edu (M.A.N.); simona.percec@temple.edu (S.P.); 2Physics Department, College of Science and Technology, Temple University, 1925 N 12th Street, Philadelphia, PA 19122, USA; 3Mechanical Engineering Department, College of Engineering, Temple University, 1947 N 12th Street, Philadelphia, PA 19122, USA; natnaeltadmassu@gmail.com (N.A.); omar.elfishi@temple.edu (O.B.E.)

**Keywords:** impact resistance, armor panel, multilayer polymer constructs, Kevlar^®^, ultrahigh molecular weight polyethylene, polyvinyl butyral, backside deformation, absorption energy

## Abstract

The impact resistance of layered polymer structures using polyvinyl butyral (PVB) in combination with Kevlar^®^ fabric and ultra-high molecular weight polyethylene (UHMWPE) were fabricated and tested. Methods of wet impregnation and hot-press impregnation and consolidation of fabric with PVB and UHMWPE were used to manufacture multilayer constructs. All sandwich constructs were fixed to the surface of ballistic clay and subject to a free drop-weight test with a conical impactor having a small contact area. All tests were made at the same impact energy of 9.3 J and velocity of 2.85 m/s. The change in the resistance force was recorded using a piezoelectric force sensor at the time intervals of 40 μs. Using experimental force–time history, the change in the impactor’s velocity, the depth of impactor penetration, the energy transformation at various stages of impactor interaction with the sample, and other parameters were obtained. Three indicators were considered as the main criteria for the effectiveness of a sample’s resistance to impact: (1) minimum deformation, bulging, of the panel backside at the moment of impact, (2) minimum absorption of impact energy per areal density, and (3) minimal or, better yet, no destruction of structural integrity. Under the tested conditions, the rigid Kevlar–PVB–Kevlar sandwich at the frontside and relatively soft but flexible UHMWPE–Kevlar–UHMWPE layers in the middle helped to localize and absorb impact energy, while the backside Kevlar–PVB–Kevlar sandwich minimized local bulging providing the best overall performance. The front layer damage area was very shallow and less than two impactor tip diameters. The backside bulging was also less than in any other tested configurations.

## 1. Introduction

Reducing the weight of protective panels of body armor and other defensive structures while maintaining their performance properties under repeated exposure to projectiles and impact from sharp objects is a priority research goal for their improvement. Currently, there are four main materials used in such panels including steel, ceramics, and polymers like polyethylene with ultrahigh molecular weight (UHMWPE), high density (HDPE), poly(p-phenylene terephthalamide) (PPTA), and combination of them. Each material has advantages and disadvantages. Historically, steel was the first reliable protective material. Its strengths are high toughness, stopping power, and high multi-hit capability. Its main disadvantage is high weight. In addition, steel deflects a large mass of shrapnel rather than absorbing the impact. Despite this, research efforts to improve the protective properties of steel and other metal alloys [1,2,3], including metamaterials [4] and metallic glasses [5], are underway. Ceramics have much higher stiffness and fracture strength and can decelerate a projectile rapidly. At the same time, these properties lead to material brittleness. Brittle behavior also means that structural cracks extend far from the point of direct impact. Even when the ceramic panel is not completely destroyed, its resistance to the next strike is significantly reduced [6]. In addition, ceramics are at least twice as heavy as polymers.

Among polymers, UHMWPE and PPTA are currently most often used as light armor protection, separately or together [7,8,9,10]. UHMWPE with a high level of crystallinity, despite a relatively low elastic modulus, has a very high fracture strain and, as a result, moderate toughness that allows to catch a bullet and localizes the effect of impact. However, the disadvantage of UHMWPE is its low melting point, which leads to its severe softening and deformation due to ballistic impacts. Its impact resistance allows it to provide protection only up to level II according to the American National Institute of Justice (NIJ) [11]. To obtain a higher level of protection, the thickness of the panel must be increased significantly, making the body armor uncomfortable to wear. At the same time, PPTA in the form of fiber and fabric, such as Kevlar^®^ from DuPont, Wilmington, DE, USA [12], Twaron^®^ from Teijin Aramid, Arnhem, The Netherland, and other aramids, is mainly used as a soft protective armor. It has a very high strength-to-weight ratio, high stiffness, significantly higher thermal stability than UHMWPE and, as a result, fairly high energy absorption [13]. The main disadvantages of the PPTA fibers are their low elasticity due to very high degree of crystallinity. Also, manufacturing of PPTA fibers is environmentally very unfriendly due to energy-intensive processes and use of large quantities of concentrated sulfuric acid. Minimizing its use would be very desirable. Moreover, PPTA fibers have a relatively short lifespan due to sensitivity to hydrolysis and ultraviolet (UV) irradiation [14,15,16]. These characteristics make it necessary to search for methods and coating materials which could alleviate such issues [16,17].

There are other polymers, fibers, and composites based on them that are analyzed for use in impact-resistive protection for warfare and also for the aerospace industry. Recently, several detailed reviews [18,19,20,21] have been devoted to the analysis of such materials and their hybridization in multilayer structures. However, it seems strange that the well-known thermoplastic polyvinyl butyral (PVB) has not been thoroughly analyzed for use in ballistic composites despite it successfully use as a bonding layer in laminated safety glass for automotive and architectural windows, preventing the crack propagation and the scattering of fragments in the event of an impact, and its good performance across wide strain rates and temperature ranges [22,23]. PVB has excellent adhesive properties, flows easily, and exhibits fast release and low solvent retention. This helps in the impregnation of yarns and fabrics, and the formation of thin film coatings [24]. Among other things, this polymer has good UV resistance and can extend the shelf life and use of PPTA by protecting it from external factors. 

However, there are some examples of PVB research for use in light armor. The work conducted by Salman’s group [25,26] used 12 w% PVB-phenolic to coat Heracron^®^ (Seoul, Republic of Korea) aramid fibers and thin films of PVB were used between woven kenaf (*Hibiscus cannabinus* L.) fibers to create a multilayer polymer composite. However, the goal of that research was focused on a potential replacement of aramid fibers with less expensive and environmentally friendly natural fibers. They studied the effect of stacking order and number of layers of kenaf and aramid on resistance to quasi-static [26] and ballistic impacts [25]. The results showed that, regardless of the order of layers, the best resistance was demonstrated by the composite without the use of kenaf. Several papers also reported the successful blending and chemical bonding of hydroxyl groups of PVB with phenolic and vinyl ester thermosets in combination with layers of aramid fabric reinforcements [27,28]. They reported that the absorption of impact energy by PVB-aramid composite is significantly higher than that of thermoset-aramid composites, but the former has a slightly greater deflection (trauma depth) due to its elasticity. The best results for both parameters, maximum energy absorption and minimum deflection, were shown by aramid composites made using a blend of PVB and thermosets. Another very recent work constructed a Kevlar^®^ composite multilayer structure from the fabric layers pre-impregnated with 50/50 mixture of phenol formaldehyde and polyvinyl butyral resins [29]. They carried out torsion, delamination, compression, and ballistic tests, but focused on constructing and comparing mesoscale and continuum numerical models and comparing them with experimental results. PVB has also been used as a matrix in 3D woven jute yarn auxetic composites [30], combined with liquid crystal polyester and poly(ester-amide) matrix resins for Twaron™ ballistic aramid fabric [31], and again, blended into the composite matrix with an unsaturated polyester thermoset resin reinforced with woven glass fabric [32]. Thus, as far as we could find, PVB in ballistic composites was mainly used as an elastic additive to brittle thermosets to improve their impact resistance. Even so, there is very little systematic analysis.

In this work, we report for the first time the comparative analysis of all-polymer multilayer composites made with Kevlar^®^49 fabric, UHMWPE film, and PVB in different combinations. To understand and optimize ballistic properties, our study first tested the tensile behavior of PVB and UHMWPE films and then examined several different combinations of them with Kevlar fabric using the free drop-weight method [33]. The behavior of a material at relatively low impact velocities does not completely determine its behavior at higher velocities, so we are currently conducting ballistic tests and the results will be reported elsewhere.

## 2. Materials and Methods

### 2.1. Components and Chemicals

The multilayer structures fabricated and tested in this work consist of various combinations of components: the para-aramid plain fabric made from DuPont™ Kevlar^®^ 49 (area density: 218 g/m^2^, thickness: 0.37 mm, plain weave, tow width and spacing 1.5 mm), UHMWPE film (molecular weight (3 − 5) × 10^6^ g/mol, density: 0.94 g/cm^3^, 0.2 mm thick) both purchased from GoodFellow, UK, and polyvinyl butyral film of two different thicknesses 0.25 and 0.05 mm (Mowital^®^ donated by Kuraray, Hattersheim, Germany [34]). Also, the following chemicals were used as received: polyvinyl butyral powder (molecular weight 67.09 g/mol, Sigma-Aldrich), acetone (ACS grade), and methanol (ultrapure, HPLC grade, Fisher Scientific).

### 2.2. Fabrication of Multilayers Structures

Kevlar^®^ fabric, sheets of UHMWPE, and PVB films were cut in shape, 6″ × 6″, using the laser cutting machine (OMTech Laser, Anaheim, CA, USA, model SH-G350, 50 W CO_2_ laser). Some of the multilayer structures were made with pieces of UHMWPE film treated with oxygen plasma on each side for one minute in the Herrick Plasma Cleaner (PDC-32, Harrick Plasma Inc., Ithaca, NY, USA), using RF high-level and 0.8 Torr of oxygen pressure. Other structures were prepared with untreated UHMWPE films. Kevlar^®^ fabric was cleaned in acetone, dipped in, and shaken for about 2 min, while PVB films were used as received.

In the manufacture of layered structures from their constituent parts, two main methods have been used (Figure 1): wet impregnation of Kevlar^®^ fabric with PVB-methanol solution and/or melt impregnation with PVB and/or UHMWPE films. In the first case, thermal consolidation with the remaining layers was carried out at the next step. In the second case, the impregnation and consolidation of all layers occurred simultaneously under the heat press. Wet impregnation was performed in the following way. PVB (10 g) powder (Sigma-Aldrich) was mixed and sonicated in 100 mL of methanol for 20 min to produce a completely transparent solution. A rectangular Kevlar^®^ fabric was submerged in the methanol/PVB 10 wt% solution for three hours while being covered. The fabric was then removed from the methanol/PVB solution and left flat to dry overnight. Melt impregnation and consolidation of all layers were achieved using an air-operated automatic heat press with top and bottom heating plates and pressure regulator (DK20SP Digital Knight 16″ × 20″ Air Automatic Heat Press, GEO Knight & Co., Inc., Brockton, MA, USA). Heat pressing was carried out at 190 °C. 

### 2.3. Tensile Measurements

Samples were laser cut in the form of standard dumbbell-shape following the ASTM D638 [35] requirements for plastic specimen thinner than 4 mm (type IV). The widest part of the sample was covered with and glued on both sides to a thick textured paper (cartridge paper) so that the flat serrated grips of the tensile machine would not pierce the polymer samples, ensuring there was no slippage and no deformation of this area during the test. Tinius Olsen model H5KT (Horsham, PA, USA) Universal Testing Machine (UTM) was used for unidirectional tensile measurements.

### 2.4. Drop-Weight Impact Device

Figure 1 shows the home-built drop-weight tower and its main components. There is a pulley at the top of the tower with the rope holding the electromagnet. This setup controls the height of the drop of and impactor assembly. The impactor assembly consists of a solenoid armature (90.4 g), additionally loaded mass, aluminum rod (20 mm diameter, 637.7 g), force sensor (26.6 g), and impactor (see Figure 1). The solenoid armature, machined from low carbon steel 1018, is the tail of the impactor assembly. The current through the solenoid (DC 24V, 10A, open frame 80N pull type electromagnet, JF-1683-24V) allows the impactor assembly to be held and released by computer control power supply. In this work, all measurements were carried out using a single impactor machined from O1 tool steel (Rockwell B95) having the dimensions indicated in Figure 1 and the mass 80.2 g. 

The quartz piezoelectric ICP^®^ force sensor (model 208C05, PCB Piezotronics, Depew, NY, USA, 22 kN range, 0.246 mV/N sensitivity, 1.05 N/mm stiffness, 36 kHz upper frequency limit, with a NIST traceable calibration) was mounted as an integrated link between aluminum rod and the impactor. This sensor is a few orders stiffer than samples under the test, thus the absorbed impact energy by the sensor is neglected. The shortest rise time of the sensor is approximately 6.9 μs, based on its nature frequency, so a recording frequency of 25 kHz and the corresponding time resolution 40 μs was chosen, which guarantees untruncated signal recording.

The specimen clamp assembly consists of two parallel aluminum plates with a 102 mm (4″) diameter hole in the center of each. The flat surface of both plates, support and top, where the test sample was placed and clamped, is 152 × 152 mm^2^ (6 × 6 in^2^). The applied compression by screws prevented slippage of the specimen in the clamp during the impact. After testing, each sample was visually inspected of the grip zone and impact zone. All measurements were carried out under conditions that the sample lies flat and gap-free on the clay (Sargent Art^®^ Plastilina, non-hardening modeling clay (Sargent Art Inc., Hazleton, PA, USA)) surface. A stainless-steel jar (108 mm (4.25 in) in diameter and 140 mm (5.7 in) depth) was filled with clay and placed in a vacuum oven at approximately 60 °C for 48 h. The jar was taken out several times and the clay were compacted to ensure full filling and no air pockets. An extra amount of clay was added to the top surface of the jar after the clay container was bolted to the bottom of the base plate (see Figure 1) to ensure the clay surface was flush with the top surface of the base plate. After each measurement, the clay was leveled with a spatula, and its temperature was checked. All measurements were performed at 21.5 ± 0.5 °C laboratory temperature. Impact tests on clay without a sample were carried out on each measurement day at its beginning and end. Separately, a series of measurements of the effect of temperature on the impact resistance of the clay was conducted at the same velocities and impact energies as in the study of our materials. These results will be published separately. 

The synchronized data acquisition system operates the following way. The rocker switch on the home-built power supply activates the relay and turns off the power to the solenoid, which initially holds the impactor at a fixed height. The same “release signal” is sent to the data acquisition system (MATLAB ver.9.1) to start collecting data points from the force sensor, and a secondary signal is sent to the Phantom camera to start recording the drop. 

All measurements are recorded in the parameters of force–time, while the most important interpretation of the results is carried out in the parameters force–displacement. Therefore, the correct transformation of the measured data into important outcome parameters, such as stiffness, toughness, energy, etc., for the understanding, modeling, and development of new materials is necessary. This often requires the use of independent methods to obtain and verify the same final parameters.

The first parameter needed for quantitative analysis is the impactor velocity, *v*_0_, when it touches the specimen. The theoretical velocity was calculated based on the energy conservation principle. The potential energy must be equal to the kinetic energy: (1)mgh=12mv2, v0=2gh.

With the drop heights equal 422 ± 1 mm, and 9.802 m/s^2^ gravitational acceleration at the location of our lab, the impact velocity, *v*_0_, is 2.877 ± 0.003 m/s. These calculations do not take into account any types of friction, including air drag, that are always present in the system. 

Since the sensor is a part of the impactor assembly and is very sensitive to acceleration, it was possible to determine the moment of separation of the impactor from the electromagnet based on its readings. Thus, by knowing the time of its fall until the moment of contact with the specimen and assuming zero initial velocity, one can calculate the impact velocity (*v* = *g* × *t*) from the force-sensor output. However, while the armature slides inside the solenoid, the force of friction plus the force of electromagnetic induction, due to the finite turn-off time of the solenoid, violates the law of free fall and acceleration is not equal to gravitational. The sensor signal recording is shown in the graph (Figure 2), where over the time interval (*t*_1_ − *t*_2_) the acceleration of the impactor increases. This makes it difficult to accurately determine impact velocity using this method. Estimates based on the time interval (*t*_3_ − *t*_1_) and (*t*_3_ – *t*_2_) give the impact velocity 2.98 m/s and 2.71 m/s, respectively. The average of these two values gives us 2.847, which agrees well with the above-calculated value of 2.877 m/s as an upper limit obtained from the principle of energy conservation. Considering that the mass of the impact assembly is 2297 g and the impact speed is 2.847 m/s, the impact energy is equal to 9.31 J.

Processing of drop-weight impact results was based on the methodology described in the ASTM standards [33,36]. Briefly, the direct measurements of the force versus time by the force sensor are converted to the force versus displacement data using the following transformations:(2)$$v\left(t\right)=v_{0}-\frac{1}{m}\;\int_{0}^{t}F_{exp}\left(t\right)\;dt$$
(3)$$s\left(t\right)=\int_{0}^{t}v_{t}\left(t\right)dt$$
(4)$$E\left(x\right)=\int_{i}^{j}F\left(x\right)dx$$
(5)$$E\left(t\right)=\frac{1}{2}m\left[v_{0}^{2}-{v(t)}^{2}\right]+mg\cdot s\left(t\right)=E_{a}+E_{e}$$
where *v_0_* is the impactor velocity at the touch moment, m is the mass of an impactor assembly, *F_exp_* is force-sensor measurement output, and *s(t)* is the impactor displacement. The energy balance at any given time, *E(t)*, or given position, *E(x)*, and the energy transfer from the impactor to the specimen can be calculated either by the integration of the force–displacement curve, Equation (4), or by the energy balance, Equation (5), where *E_a_* is the absorbed energy and *E_e_* is the elastic energy. Here, *i* and *j* are any two points along the displacement curve. As a reference, all integration is conducted using the midpoint rule. The steps of such transformation are illustrated in Figure 3. In this way, all parameters of the impact resistance of the sample including the energy of absorption at different phases of impactor-specimen interaction can be calculated.

## 3. Results

### 3.1. Unidirectional Tensile Measurements

The construct of the studied multilayer composite structure is based on the components purchased from various sources listed above. It should be noted that the parameter values specified by the manufacturer are not always precise technical characteristics of the received materials. One example could be given about the characteristics of UHMWPE film from GoodFellow, where according to the product specifications, the tensile modulus is 0.2–1.2 GPa, tensile strength 20–40 MPa, and elongation at break 500%. As one can see, this covers a wide range of values, especially for tensile modulus, and cannot be used to analyze and understand the behavior of the multilayer composite structures. During the drop-weight impact tests, it turned out that the UHMWPE films also have a certain level of anisotropy with respect to the direction of the roll. To obtain precise parameters of the film, it was cut in two orthogonal directions relative to the twist of the roll. Tensile measurements of these samples were performed using different strain rates. Below are two figures representing the stress vs. strain behavior for the case when the samples were cut across the direction of the roll (Figure 4b) and along the direction of the roll twist (Figure 4c). Tensile measurements of PVM Mowital® 0.25 mm thick films were also performed at three strain rates (Figure 4e), the same as for UHMWPE. The measured parameters, and the secant modulus of elasticity, *E_s_*, calculated at 3% tensile strain and ultimate tensile strength (UTS) for UHMWPE and PVB films obtained at different loading rates and load directions, are presented in Table 3. 

### 3.2. Drop-Weight Impact Test 

Using a force–displacement history diagram (Figure 5), the stages of impact energy transfer to a composite specimen and how it behaves at each corresponding stage can be described. For comparison, this is illustrated using four samples representing three basic types of behavior shown in Figure 5a–c. In the first stage, the load curve moves from the touch point to the first bend relatively slowly. According to [36], this stage can be caused by the inertial effects in the striker and, in our case, is most likely the result of sample displacement with the low resistance due to its initial bulging, minor gaps, and irregularities remaining after it was placed on the clay baking surface and clamped. This is denominated as the preloading zone (PL). For stiff samples, this zone is significantly shorter and almost vanishes (e.g., Figure 5b). After this, the force–displacement diagram becomes steeper and approximately linear until the first crack or delamination appears (*Fc*, Figure 5b), or force approaches its peak or maximum value (*Fm*, Figure 5a). The slope of this curve is a measure of the system's elastic response and its relative stiffness (RS) [36]. In our case, when the sample lies on the clay baking surface, this parameter reflects both the order of transverse deformation and the magnitude of its longitudinal distribution. A detailed analysis of the experimental results will involve a three-dimensional reconstruction of the profile of the deformed sample and the response of the clay basin [37]. A quantitative comparison of this parameter for different samples requires considering the thickness of samples and the same experimental conditions. 

In the next zone, features of the load–displacement behavior above *F_c_* and around the peak load *F_m_* reflect the type of deformation and the level of sample damage, if any. This can represent different failure modes—irreversible deformation without or with cracks, delamination, or complete sample fracture. In general, this is called the failure zone (FZ). None of the tested samples were pierced, but samples with a combination of Kevlar^®^ and UHMWPE demonstrated irreversible deformation without any failure (Figure 5a), whereas, as a result of an impact on samples of Kevlar^®^ with PVB, but without UHMWPE, (Figure 5b) the destruction is well recorded as sudden load drops and recovery. The post-test inspection revealed ruptures of the Kevlar^®^ fibers and cracks in the PVB on the front side of the composite plate. In both extreme cases, a smooth funnel in Kevlar^®^–UHMWPE sample and a much smaller funnel but with breaks in Kevlar^®^–PVB sample, are visible clearly in optical photographs with a digital zoom (Figure 6a,b). The combination of Kevlar^®^ fabric with PVB and UHMWPE as the back layer results in a visible reduction of fracture and sagging of the front surface and reduced bulging of the back surface (Figure 6d).

Noteworthy is the fact that after the peak load, *F_m_*, there is a zone of rapid drop in force and continued penetration of the impactor (Figure 5). A likely explanation for this behavior could be a progressive sample damage beyond the maximum load [38] and the viscoelastic nature of these samples [39]. To fully understand its nature, a special set of measurements is required at various impact rates and energies. 

The peak force value, *F_m_*, corresponds to the maximum negative acceleration, or deceleration, of the penetrated impactor, whereas the maximum deformation corresponds to the moment when the penetrated impactor velocity reaches zero. After the impactor velocity changes from zero to a negative value, the impactor begins to move up together with sample, and the force continues to decrease. It corresponds to the elastic response of the target (ER) when the sample partially restores its initial thickness and position to the value of residual deformation (*R_d_*). The measured force goes to zero at the moment when impactor separates from the sample surface. It now measures only a small negative value corresponding to the inertia of the deceleration until the striker reaches the highest rebound point. After that, it again begins to move down for a second strike. The maximum value of the released elastic energy in our measurements was not more than 11%. Therefore, the energy of the second impact is insignificant, and we do not take into account that this can cause additional deformation and damage. Also, any deformation of the steel impactor assembly is neglected since the stiffness of its components is several orders of magnitude higher, and the deformation of the sensor is negligible. 

In Figure 5, the ER phase displays the release of the elastic component of the initial impact energy. As one can see, the ER of the sample with UHMWPE (Figure 5a) is more than twice that of the sample with PVB only (Figure 5b). Figure 7a shows the energy–time histories of behavior for three samples as in Figure 5 and for a thinner sample PE3K2 for contrast. In Figure 7b, the maximum deformation and the instant elastic recovery for these samples are shown for comparison. It should be noted that due to the viscoelastic nature of the composite panels, the final recovery of deformation is much greater than the immediate recovery recorded during each test. However, in this work, we do not have reliable qualitative measurements of the slow recovery process and one can only approximately measure the final state of its deformation after tens of minutes. So, for K5PE4, the final deformation is approximately 3.4 mm, and for K5PVB2PE2 it is about 2.5 mm, which is around 4.8 mm less than what was measured at the time of the test in both cases.

All the main results of the experimental data and their numerical analysis are given in Table 4. Also presented here are the maximum stresses arising on the sample surface at the peak impact force, considering the impactor flat tip with a diameter of 1.17 mm (Figure 1h) and a corresponding contact area of 1.07 mm^2^.

## 4. Discussion

When comparing the behavior of different multilayer composite structures (Figure 5), the first thing that becomes evident is the significantly lower peak load, *F_m_*, for sample K5PVB3 (Figure 5b) and the accompanying wide zone and weak resistance to destruction. This is confirmed by photographs of the front and back sides (Figure 6b). On the frontside, one can see all possible damage—plastic deformation and cracking, fiber-matrix debonding, fiber fracture, and tow splitting. The backside is also deformed significantly. Tows are dislodged and some fibers are torn. This also means that the PVB layers are broken into pieces and fabrics are delaminated from polymer layers. This sample consists of five layers of Kevlar^®^ fabric, three of which are impregnated with PVB polymer, and then all five are consolidated at the pressure and temperature of 190 °C. 

The completely different force–displacement dependence is for sample K5PE4 (Figure 5a), consisting of five layers of Kevlar^®^ fabric and four layers of UHMWPE films consolidated under hot pressing at 190 °C. In this case, no polymer cracking and fiber fracture is observed at the frontside (Figure 6a). At the same time, there is a large deformation of the backside (button image of Figure 6a). This deformation is not localized, and it is distributed over a significant distance from the point of impact. For this reason, it is difficult to see it in a photograph, and two white lines are added to help guide the viewer’s eye. In addition, a large elastic component leads to a significant transfer of momentum to a protected body, which can have a negative effect too. 

The difference in behaviors of K5PE4 and K5PVB3 samples can be explained by the inherent difference in the tensile properties of UHMWPE and PVB (Figure 4). The PVB is stiffer and harder but has much less fracture strain than UHMWPE. Thus, at relatively low strain rate, their behavior is significantly different. It is known that polyethylene experiences a strain hardening at a higher loading rate [40,41], and PVB is a viscoelastic polymer with a strong temperature and load-rate dependence [22,42] too. Experiments to compare these results with ballistic measurements of the same structures are in progress and will be reported elsewhere. 

By combining these two extreme cases, a set of layered samples from the Kevlar^®^ fabric and both UHMWPE and PVB were created. The use of UHMWPE layers at the back in sample K4PVB1PE2 resulted in an increase in peak load and a decrease in its maximum deformation (Figure 5d, red dots). However, at the same time, the brittleness of the front layer was preserved, although the damage zone FZ decreased (Figure 6c). In the next set of samples, a layered structure K5PVB2PE2 was created where two layers of UHMWPE were embedded in the middle of the sandwich. This combination gives the best results in all parameters (see Table 4). Upon a very detailed examination, one can still notice two small resets in the load resistance, marked in Figure 5c as *F_c1_* and *F_c2_*, although these destructions of the front layer are local and insignificant. Such localized failure only affects an area of less than two impactor tip diameters or less than two tows of yarn. There is no evidence of fibers being pulled out beyond this area. The backside deformation in the case of K5PVB2PE2 looks very similar to K5PE4, but, in fact, its value is much less (see Table 4). Here, it must be emphasized once again that the measured residual deformation during the test is not final, primarily due to the significant viscoelasticity of UHMWPE. For this reason, after a few minutes, the permanent deformation is significantly less than at the moment of impact, as was reported here earlier. However, for the object that needs to be protected, the only important parameters are the maximum penetration depth of the impactor and the impactor energy that the armor is capable of absorbing.

By analyzing the transfer of energy from the impactor to the panel as a function of time (Figure 7a) and as a function of displacement (Figure 7b) during deformation and ultimately possible penetration of the impactor into the sample, the following conclusions can be drawn. At the initial stage, in terms of time and displacement, K5PVB3 absorbs impact energy faster than other materials (Figure 7a green curve), which indicates it has higher stiffness, compared to others. As expected, PE3K2 has the lowest stiffness (Figure 7a, b blue curve) and relatively small elastic component, *E_e_*, which indicates irreversible damage and significant plastic deformation at the applied impact energy. K5PE4 (brown curve) absorbs the most elastic energy and, accordingly, demonstrates maximum elastic deformation, which is then returned to the impactor for its rebound. Due to this, its residual deformation is relatively small and there is virtually no fabric destruction, but the main disadvantage of this structure is significant deflection and bulging upon impact (Table 4). The second most rigid structure, also as anticipated, is K5PVB2PE2 (Figure 7a,b red curves). At the same time, this structure accumulates more elastic energy than other structures except K5PE4. If we compare the stiffness and energy absorption behavior of K5PVB2PE2 and K4PVB1PE2 (it is not shown in Figure 7), the latter, which has one layer less of Kevlar^®^ and one layer less of PVB is slightly weaker and less elastic. Here, it is necessary to clarify that the total thickness of PVB in K4PVB1PE2 is 0.25 mm and in K5PVB2PE2 is 2 × 0.05 = 0.10 mm. Thus, the last sample has 2.5 times less PVB than the first has, and its area density is only slightly more due to an extra layer of Kevlar^®^ fabric. (see Table 1 and Table 2). So, the weaker behavior of K4PVB1PE2 is not surprising. The most important thing is that this causes very significant structural damage, although it is quite localized (Figure 6c). There is also a relatively large bulging of the backside without cracking and fracture. Thus, based on the analysis of all parameters, it can be concluded that the structural combination K5PVB2PE2 performs the best, including protecting against multiple impacts.

## 5. Conclusions

From the entire analysis above, it should be concluded that the advantages of only one or two properties are not yet sufficient for the overall acceptable behavior of the armor structure. The goal of this work was to optimize the design of a sandwich structure made of Kevlar^®^ fabric, UHMWPE, and PVB to obtain the best combination of three parameters: (1) minimum deformation of the backside of the panel at the moment of impact, (2) minimum absorption of impact energy per areal density, and (3) minimal local destruction of structural integrity. Using Kevlar^®^ fabric (K), UHMWPE (PE), and PVB in the sequence K–PVB–K–PE–K–PE–K–PVB–K demonstrated the best results. Impregnation of Kevlar^®^ fabric with PVB, prevents fibers, yarns, and tows from being pulled out from the woven structure. At the same time, a large number of internal interfaces between PVB and Kevlar fibers prevents the cracks propagation inside the binder polymer. This localizes the irreversible destruction of the composite structure. Several layers of less rigid, but not brittle, easily deformable, and tough UHMWPE in combination with the fabric behind the K–PVB–K sandwich damped the impact, preventing the propagation of brittle failure and absorbing out-of-plane structural deformations. By supplementing the multilayer structure with a sandwich made of K–PVB–K, the same as the front side, we obtained minimal deformation and bulging towards the protected body. 

Optimization of such multilayer structures, as well as their interfacial properties, for specific impact energies and impact velocities, is the goal of our current and future research. Based on the logic of the obtained results, thicker layered structures will be tested for ballistic resistance. 

## Data Availability

The raw data supporting the conclusions of this article will be made available by the authors on request.

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
