# Peer review of "Damage Resistance of Kevlar® Fabric, UHMWPE, PVB Multilayers Subjected to Concentrated Drop-Weight Impact"

_polymers, 2024, doi:10.3390/polym16121693_

Round 1
Reviewer 1 Report
Comments and Suggestions for Authors
This is a good paper with novel results and just a few minor points need to be addressed prior to publication:
1. It is a well-written paper with a very clear and good standard of English. Just one phrase needs re-writing please: Line 60 - ‘Its impact resistance allows to cover only…’
2. Please add a colour code for the different layers to Table 1 (continuation) to help the reader.
3. Line 285, etc: - do you really wish to give impact velocities to an accuracy of three figures after the decimal point ?
4. Line 342: Fc in italics ?
5. Line 343: ‘peak load, Fm,’ ?
6. Line 404: Fm in italics ?
Reviewer 2 Report
Comments and Suggestions for Authors
The manuscript “Damage Resistance of Kevlar® Fabric, UHMWPE, PVB Multi-layers Subjected to Concentrated Drop-Weight Impact aims to study the impact resistance of a composite made of Kevlar, UHMWPE, and PVB with different architectures. Tensile tests and their impact resistances obtained the samples' mechanical properties were measured by drop test.
The results are presented and conclusions are drawn.
The abstract does not clearly state the authors' methodology, results, and conclusion.
What is the scientific contribution of this work? Explain in the introduction section.
What is the applicability of the newly developed material? Explain in the introduction section.
Equations from pages 7 and 8 must be in the Drop-weight impact device section.
The results and discussion sections do not contain results already studied on the same or similar material. It is necessary to compare the results with those found in the literature.
In section 2, How many samples were used for each mechanical test?
Section 3 presents a table with the maximum average values and the respective standard deviations for the sample and test.
Page 3 line 123, correct the unit “Mg/mol”.
Page 4, Figure 6: Indicate the damaged area and its depth when applicable. Improve image quality.
The Conclusions section is very long. Authors should divide it into numbered gaps. Emphasise the main novelty and advantages of the results obtained. Separate the detailed results against those general in each of the numbered gaps. Provide any further outline to proceed with the study in the future.
Reviewer 3 Report
Comments and Suggestions for Authors
The author has prepared a layered composite structure consisting of PPTA fabric, UHMWPE, and PVB using wet and hot-press impregnation methods with the aim of investigating its impact resistance using a free-falling drop-weight conical impactor with a quasi-static load, a low impact rate load, and a fast projectile. While the research paper demonstrates a scientifically important area of research, several points must be addressed before the manuscript is accepted for publication in the polymers journal, which are listed as follows:
- In the abstract, it is recommended to detail the key findings related to composite impact behavior under different mechanical load conditions.
- The elaboration on the metals and ceramic materials is of no use in the introduction part as the study focuses on composite systems fabricated mainly from polymers.
- It is rather suggested to strengthen the introduction by providing a review of the development and optimization of the ballistic protection properties of polymer composites.
- It is recommended to provide a schematic representation to clearly illustrate all steps used in wet and hot-press impregnation processes to fabricate the composite material under the “materials & methodology” section.
- Introduce (a) in Figure 1 at the beginning of the sentence.
- In formula (1), the velocity v0, may be perceived as the initial velocity. It's better to use another symbol.
- The kinematic equation in Line# 279, can’t be used in calculation as the force displays variation, and hence the energy approach indicated by the conservation of energy principle is the appropriate alternative.
- For the Figure 4 caption, introduce (a), (b), (c),... at the beginning of the sentence. This has also been repeated in other figures.
- Does the force sensor record the force variation if the sample falls from a certain height, as indicated by 422 mm?
- In Figure 4 (b), does the positive velocity value when t < 0.01 s represent the rebound of the impactor? Clarify this.
- Line# 356, Figure 6 (a-d) is claimed to represent the multi-layered structure composites of Kevlar, PVB, and UHMWPE; do all figures (a-d) represent that?
- Figure 6 represents four samples, but in the caption, three were mentioned. Clarify this.

Comments on the Quality of English LanguageMinor editing and formatting is required, as highlighted in the manuscript.
Round 2
Reviewer 2 Report
Comments and Suggestions for Authors
The manuscript “Damage Resistance of Kevlar® Fabric, UHMWPE, PVB Multi-layers Subjected to Concentrated Drop-Weight Impact aims to study the impact resistance of a composite made of Kevlar, UHMWPE, and PVB with different architectures. Tensile tests and their impact resistances obtained the samples' mechanical properties were measured by drop test.
After the revision, this paper could be accepted because the authors corrected all items indicated by this reviewer.